# Nutrigenomic Studies on the Ameliorative Effect of Enzyme-Digested Phycocyanin in Alzheimer’s Disease Model Mice

**DOI:** 10.3390/nu13124431

**Published:** 2021-12-10

**Authors:** Yasuyuki Imai, Yurino Koseki, Makoto Hirano, Shin Nakamura

**Affiliations:** 1Health Care Technical G., Chiba Plants, DIC Corporation, Ichihara 290-8585, Chiba, Japan; yasuyuki-imai@ma.dic.co.jp (Y.I.); yurino-koseki@ma.dic.co.jp (Y.K.); 2R&D Institute, Intelligence & Technology Lab, Inc., Kaizu 503-0628, Gifu, Japan; mhirano@itechlab.co.jp; 3Biomedical Institute, NPO Primate Agora, Kaizu 503-0628, Gifu, Japan

**Keywords:** Y maze, *Spirulina*, Creutzfeldt–Jakob disease, neurodegenerative disease, hippocampus, amyloid β-peptide(_25–35_)

## Abstract

Alzheimer’s disease (AD) is the most common form of dementia, and the cognitive impairments associated with this degenerative disease seriously affect daily life. Nutraceuticals for the prevention or delay of AD are urgently needed. It has been increasingly observed that phycocyanin (PC) exerts neuroprotective effects. AD model mice intracerebroventricularly injected with amyloid beta-peptide 25–35 (Aβ_25–35_) at 10 nmol/head displayed significant cognitive impairment in the spontaneous alternation test. Cognitive impairment was significantly ameliorated in mice treated with 750 mg/kg of enzyme-digested (ED) PC by daily oral administration for 22 consecutive days. Application of DNA microarray data on hippocampal gene expression to nutrigenomics studies revealed that oral EDPC counteracted the aberrant expression of 35 genes, including *Prnp*, *Cct4*, *Vegfd* (*Figf*), *Map9* (*Mtap9*), *Pik3cg*, *Zfand5*, *Endog*, and *Hbq1a*. These results suggest that oral administration of EDPC ameliorated cognitive impairment in AD model mice by maintaining and/or restoring normal gene expression patterns in the hippocampus.

## 1. Introduction

Alzheimer’s disease (AD) is the most common form of dementia. The symptoms of AD include memory loss and mild cognitive impairment at onset, but as the disease progresses, some patients suffer from severe dementia, ultimately leading to death in extreme cases. According to Alzheimer’s Disease International, dementia affects over 50 million people worldwide, and a significant portion of the cases are estimated to be caused by AD. As the aging population increases, the number of AD patients is also considerably increasing.

AD is characterized by amyloid β (Aβ) plaques and neurofibrillary tangles of tau protein [1]. One of the two AD subtypes, late-onset sporadic AD, occurs after 65 years of age. The etiology of this more common form of AD has not been fully elucidated, but the amyloid hypothesis is widely accepted. According to this theory, neurotoxic peptides derived from amyloid precursor protein (APP), typically Aβ_1–42_, play pivotal roles in AD etiopathology. Rodents intracerebroventricularly (i.c.v.) injected with Aβ_1–42_ or Aβ_25–35_, the most neurotoxic part of Aβ_1–42_, are widely used for fundamental research on AD pathology and for developmental research on the prevention and treatment of AD [2,3].

Biogen Inc. (Cambridge, MA, USA) and Eisai Co., Ltd. (Tokyo, Japan). have developed a humanized anti-Aβ monoclonal antibody, which is called aducanumab, for the treatment of AD. In June 2021, the US Food and Drug Administration (FDA) granted accelerated approval for aducanumab. Since the first description of this disease in 1906, aducanumab is the first therapeutic drug to be approved for AD treatment by targeting the causative agents rather than the symptoms. However, approval by the FDA did not settle the debate over the efficacy of aducanumab. The demand for nutraceuticals or functional foods for the prevention or delay of AD is thus still prevailing.

A group of blue-green algae of the genus *Arthrospira* (formerly *Spirulina*) utilizes the photosynthetic protein phycocyanin (PC), which is blue, as its name suggests. It has been increasingly found that the consumption of *Spirulina* or PC extracts is associated with a variety of beneficial effects, such as antioxidant, anti-inflammatory, immunomodulatory, hepatoprotective, nephroprotective, and neuroprotective effects [4,5]. Oral administration of PC ameliorated the neurotoxic effects of kainic acid in rats [6]. Intraperitoneal administration of PC alleviated cognitive deficits in rats i.c.v. injected with streptozotocin (STZ) [7]. Li et al. described that i.c.v. injection of Aβ_1–42_ impaired spatial memory performance in mice. These cognitive deficits were ameliorated by intraperitoneal administration of PC [8]. The hippocampus is indispensable for cognitive functions, including memory, and is one of the most severely affected parts of the brain in AD patients [9]. In search of AD biomarkers, Hosseinian et al. conducted a meta-analysis on aberrantly expressed genes in the hippocampus [10].

In this study, i.c.v. injection of Aβ_25–35_ altered the gene expression profile of the hippocampus and resulted in cognitive impairment in mice. Our experimental results suggested that, of orally administered PC products (PC and EDPC), EDPC prevented Aβ-induced transcriptional changes in the hippocampus, resulting in an alleviation of the associated cognitive impairment in vivo.

## 2. Materials and Methods

### 2.1. PC Products and Aβ_25–35_

*Spirulina* was a dried product of *Arthrospira platensis*, which was obtained from DIC Co., Ltd. (Tokyo, Japan). ED *Spirulina* was prepared by proteolysis using Protin SD-NY10^®^ from Amano Enzyme Co., Ltd. (Nagoya, Japan). Purified PC, called PC, was purified from water extract of *Spirulina* by filtration followed by ultrafiltration. The EDPC employed was a proteolysis product of PC generated by Protin SD-NY10^®^-mediated cleavage, in which no less than 58% were low-molecular-weight components with a molecular weight less than 6 kDa.

Aβ_25–35_ was purchased from Peptide Institute, Inc. (Osaka, Japan). Distilled water for injection was purchased from Otsuka Pharmaceutical Co., Ltd. (Tokyo, Japan). Prior to use, Aβ_25–35_ was dissolved in distilled water and incubated at 37 °C for 4 days to promote aggregation.

### 2.2. Animals

Four-week-old male Slc:ddY SPF mice were purchased from Japan SLC Co., Ltd. (Shizuoka, Japan). The mice were housed in polycarbonate plastic cages under a 12 h light/12 h dark cycle with free access to tap water and a standard rodent diet of MF chow (Oriental Yeast Co., Ltd., Tokyo, Japan). After one week of acclimation (i.e., at 5 weeks of age), the mice were divided into seven groups (A1 to A7; 4–6 mice/group). The average body weight (BW) of the mice was essentially identical among all groups during the duration of the experiment.

At 5 weeks of age, *Spirulina* (750 mg/kg BW), ED *Spirulina* (750 mg/kg BW), PC (750 mg/kg and 120 mg/kg BW), and EDPC (750 mg/kg BW) were administered orally once daily for 22 consecutive days. Control individuals (groups A1 and A2) received distilled water (Table 1). At 7 weeks of age (14 days after the start of the treatment), the mice were anesthetized with 75 mg/kg ketamine and 10 mg/kg medetomidine, and each mouse was fixed on an SN-1 stereotaxic apparatus (Narishige International Ltd., Tokyo, Japan). Aβ_25–35_ aggregates were i.c.v. injected at 10 nmol/mouse using a 50 μL microsyringe fitted with a 27 gauge needle. Control individuals (group A1) were injected with distilled water. At 8 weeks of age (7 days after i.c.v. injection of Aβ_25–35_), spontaneous alternation tests were performed using a Y maze, after which the mice were euthanized by decapitation (Figure 1).

The study protocol was approved by the Animal Care Committee of Intelligence and Technology Lab, Inc. (ITL). Experiments were performed in accordance with the Institutional Animal Care and Committee Guide of the ITL based on the Guidelines for Proper Conduct of Animal Experiments.

### 2.3. Y Maze Spontaneous Alternation Test

When moving from one place to another, rodents exhibit a natural tendency to explore the least visited area. This behavior is referred to as spontaneous alternation. Spontaneous alternation in a Y maze is used to evaluate cognitive function, especially short-term spatial recognition memory [11]. The Y maze consisted of black-painted plywood, with three equal arms (40 cm long, 12 cm high, 3 cm wide at the bottom, and 10 cm wide at the top) oriented at 120° angles to each other. Each mouse was placed at the end of an arm and allowed to explore the maze for 8 min. A session with a total number of arm entries (N) ≤ 8 was considered to indicate defects in locomotor activity, and the test result was consequently excluded from the analysis. If a mouse entered all three arms in succession without repetition, this was defined as alternation behavior. For example, if the three arms were called A, B, and C, the number of alternations (M) was four in the case of ABCBACACB. The spontaneous alternation ratio (R) was calculated according to the formula: R (%) = M × 100/(N − 2).

### 2.4. Total RNA Isolation

The hippocampi of the mice in groups A1, A2, A5, and A7 were harvested after performing the spontaneous alternation test. Tissues from these four groups were submerged in RNAlater (Thermo Fisher Scientific K.K., Tokyo, Japan) and stored at −70 °C. Four hippocampi (taken from two mice) per group were pooled and homogenized in RNAiso Plus (Takara Bio Inc., Shiga, Japan). The RNA extracted into the aqueous phase was treated with DNase (QIAGEN K.K., Tokyo, Japan) and further purified using the RNeasy MinElute Cleanup Kit (QIAGEN) according to the manufacturer’s instructions. Absorbance was measured at 230, 260, 280, and 320 nm using an Ultrospec 2000 (Pfizer Japan Inc., Tokyo, Japan), and the RNA integrity number (RIN) was determined using an Agilent 2100 Bioanalyzer (Agilent Technologies Japan Ltd., Tokyo, Japan).

Only high-quality RNA samples, which were defined by an A_260_/A_230_ ≥ 1.5, A_260_/A_280_ ≥ 1.8, and RIN ≥ 6.0, were used in downstream microarray analyses.

### 2.5. DNA Microarray Analysis

DNA microarray analyses were performed for the four groups, i.e., A1, A2, A5, and A7. After cDNA synthesis, Cy3-labeled cRNA was synthesized and purified using the Low Input Quick Amp Labeling Kit (Agilent) according to the manufacturer’s instructions. Notably, reverse transcription was conducted using a T7 promoter-oligo(dT) primer. Absorbance was measured at 260, 280, 320, and 550 nm, and it was verified that the labeled cRNA had incorporated >6 pmol/mg of Cy3-CTP. Labeled cRNA was fragmented using the Gene Expression Hybridization Kit (Agilent) and applied to Whole Mouse Genome Array Ver2.0 slides (Agilent). After hybridization at 65 °C for 17 h, the slides were washed with Gene Expression Wash Buffers 1 and 2 (Agilent) according to the manufacturer’s instructions. The slides were scanned using a GenePix 4000B scanner (Molecular Devices Japan K.K., Tokyo, Japan). Scanned images were digitalized and normalized using GebePix Pro software (Molecular Devices Japan K.K., Tokyo, Japan).

### 2.6. Bioinformatic Analysis of Microarray Data

By comparing groups A1 (sham: oral water + i.c.v. water) and A2 (Aβ: oral water + i.c.v. Aβ_25–35_), a list of genes that were >2-fold upregulated or <0.5-fold downregulated in group A2 was generated. We referred to these genes as Aβ-related genes. By comparing groups A2 and A5 (Aβ + EDPC: oral EDPC + i.c.v. Aβ_25–35_), genes relevant to the amelioration of impaired cognitive function were extracted from the list of Aβ-related genes. More specifically, if an upregulated Aβ-related gene was <0.5-fold downregulated or a downregulated Aβ-related gene was >2-fold upregulated in group A5, we considered the gene product relevant to the observed amelioration and referred to it as EDPC-related gene. Similarly, by comparing groups A2 and A7 (Aβ + PC: oral PC + i.c.v. Aβ_25–35_), we defined PC-related genes. The top 50 Aβ-related genes and all EDPC- and PC-related genes were annotated, and references were searched using NCBI databases and Google. The physiological and pathological functions of annotated genes were deduced with respect to the possible mechanisms of action of PC and/or EDPC.

### 2.7. Statistical Analysis

Student’s *t*-test or Aspin–Welch *t*-test was used for comparisons between two groups after the equality of the two variances had been examined by an *F*-test. Dunnett’s test or Steel’s test was used for multiple comparisons after the equality of variances had been examined using Bartlett’s test. All analyses were conducted using StatLight software (Yukms, Co., Ltd., Kanagawa, Japan). A *p*-value ≤ 0.05 was considered statistically significant.

## 3. Results

### 3.1. Spontaneous Alternation

The mice were treated with *Spirulina* (750 mg/kg), ED *Spirulina* (750 mg/kg), PC (750 mg/kg and 120 mg/kg), EDPC (750 mg/kg), or vehicle by daily oral administration for 22 consecutive days. One week after i.c.v. injection of Aβ_25–35_ (10 nmol/head) or vehicle (distilled water), spontaneous alternation tests were conducted (Figure 1). R and N values, which serve as parameters relevant to assessing cognitive function and locomotor activity, respectively, are summarized in Table 1 and Figure 2A,B. Obtained N values were not significantly different between group A1 (sham: oral water + i.c.v. water) and any other group (Figure 2A), suggesting that the locomotor activity of the mice was not significantly affected by the experimental procedures, including oral administration and i.c.v. injection. As shown in Table 1, the R value of the sham group was 75.1 ± 2.5%. That of group A2 (Aβ: oral water + i.c.v. Aβ_25–35_) was 51.0 ± 2.1%, which was significantly lower than that of the sham group (*p*-value < 0.01; Figure 2B), indicating that cognitive function was impaired in mice i.c.v. injected with Aβ_25–35_. The R value of group A5 (Aβ + EDPC: oral EDPC + i.c.v. Aβ_25–35_) was 61.8 ± 3.7%, which was higher than that of the A2 (Aβ group (*p*-value < 0.05; Figure 2B)).

These results suggest that orally administered EDPC had an alleviative effect on cognitive impairment in mice. This effect was not observed in mice treated with orally administered *Spirulina*, ED *Spirulina,* or PC (750 mg/kg and 120 mg/kg; Figure 2B).

### 3.2. Gene Expression Profiles of the Hippocampi of EDPC- and PC-Administered Mice

To elucidate the mechanism underlying the EDPC-mediated alleviation of cognitive impairment in model mice, we performed DNA microarray analyses using the hippocampi isolated from mice of sham, Aβ, Aβ + EDPC, and Aβ + PC groups (i.e., groups A1, A2, A5, and A7).

#### 3.2.1. Upregulated Aβ-Related Genes

By comparing the gene expression profiles between sham and Aβ groups, we extracted 1368 genes that were >2-fold upregulated by i.c.v. Aβ_25–35_, which are referred to as upregulated Aβ-related genes. As shown in Table 2, some of the upregulated Aβ-related genes were previously shown to be upregulated in the hippocampi of AD model mice used for microarray analyses. Thus, as our model mice displayed cognitive impairment, they were considered appropriate for examining the effect of EDPC on gene expression patterns in the hippocampus.

#### 3.2.2. Downregulated EDPC-Related Genes

We compared the gene expression profile of the Aβ + EDPC group with that of the Aβ group. Of the upregulated Aβ-related genes, three genes were <0.5-fold downregulated by oral administration of EDPC, which resulted in a significant alleviation of the cognitive impairment observed in model mice, as mentioned above. We referred to these genes as downregulated EDPC-related genes (Figure 3). As shown in Table 3, the downregulated EDPC-related genes were *Fbxl19* (F-box and leucine-rich repeat protein 19), *Pax1* (paired box gene 1), and *Zfp292* (zinc finger protein 292). FBXL19 is a ubiquitin ligase, whereas ZFP292 and PAX1 are transcription factors containing zinc fingers and a paired box domain, respectively. According to the NCBI Gene database, *Fbxl19*, *Pax1*, and *Zfp292* are highly or moderately expressed in the central nervous system (CNS) of fetal mice. However, the possible roles of these genes in the observed in vivo amelioration remain elusive.

#### 3.2.3. Downregulated PC-Related Genes

We compared the gene expression profile of the Aβ + PC group with that of the Aβ group. Of the upregulated Aβ-related genes, 16 genes were <0.5-fold downregulated by oral PC, which were referred to as downregulated PC-related genes (Figure 3). *Fbxl19*, *Pax1*, and *Zfp292* were shared with the downregulated EDPC-related genes, and the other 13 genes are summarized in Table 4. These 13 genes were also downregulated by EDPC, but the extent of the downregulation was not significant. These results suggest a common mechanism of action associated with the two PC products, i.e., PC and EDPC. Of the downregulated PC-related genes, *Abat* (4-aminobutyrate aminotransferase) and *Brp44* (brain protein 44; also known as *Mpc2*) are reportedly involved in AD (Table 4).

#### 3.2.4. Downregulated Aβ-Related Genes

By comparing the gene expression profiles between sham and Aβ groups, we identified 949 genes that were <0.5-fold downregulated by i.c.v. Aβ_25–35_. These genes were referred to as downregulated Aβ-related genes (Figure 3).

#### 3.2.5. Upregulated EDPC-Related Genes

We compared the gene expression profile of the Aβ + EDPC group to that of the Aβ group. In total, 32 of the downregulated Aβ-related genes were >2-fold upregulated by orally administered EDPC. We referred to these genes as upregulated EDPC-related genes (Figure 3). We found that *Prnp*, which encodes the etiological agent of Creutzfeldt–Jakob disease (CJD), was included in the EDPC-related genes. Seven other EDPC-related genes are known to be associated with AD or CJD (Table 5). These AD/CJD-associated genes were *Prnp* (prion protein), *Cct4* (chaperonin containing Tcp1, subunit 4), and *Figf* (c-fos-induced growth factor; also known as *Vegfd*), which are reportedly relevant to AD. Prion protein is the etiological agent of prion diseases, which is involved in the autophagy and protection of neurons under physiological conditions. CCT4 is involved in protein folding, translation, and dendrite morphogenesis, while VEGFD is involved in angiogenesis, protection of neurons, and the reconstruction of dendrites. *Olfr181* (olfactory receptor 181), *Olfr847*, *Olfr859*, and *Olfr963* were also identified as upregulated EDPC-related genes. Reportedly, certain olfactory receptors are relevant to both AD and CJD, a prion disease. *Gal3st1* (galactose-3-*O*-sulfotransferase 1, also known as *Cst*) was found to be involved in the pathogenesis of CJD.

Other upregulated EDPC- and PC-related genes, as well as the presumed functions of the gene products encoded were as follows (Table 6): *Ccdc163* (coiled-coil domain containing 163), wake; *Kcnj12* (potassium inwardly rectifying channel, subfamily J, member 12), action potential; *Lamb4* (laminin subunit beta 4), laminin and glia cells; *Mtap9* (microtubule-associated protein 9; also known as *Map9*), microtubules and cell cycle; *Pik3cg* (phosphatidylinositol-4,5-bisphosphate 3-kinase catalytic subunit gamma), microglia, induction of ER stress, and synaptic plasticity; *sema3b* (SEMA domain, immunoglobulin domain (Ig), short basic domain, secreted 3B; also known as semaphorin 3B), removal of spines of dendrites; *Tmem82* (transmembrane protein 82), cell proliferation; *Zfand5* (zinc finger, AN-1-type domain 5; also known as ZNF216), inflammation, apoptosis, and protein degradation; *Gm14439* (predicted gene 14439), ribosomal protein; *LOC100503859* (RIKEN cDNA 1110015O18 gene), unknown; *D330050I16Rik* (RIKEN cDNA D330050I16 gene), unknown.

The remaining upregulated EDPC-related genes, which were not PC-related, are shown in Table 7: *Cd244* (natural killer receptor 2B4), microglia and regulation of NK cells; *Chst9* (carbohydrate (*N*-acetylgalactosamine 4-*O*) sulfotransferase 9), modification of pro-opiomelanocortin; *Endog* (endonuclease G), apoptosis and necrosis; *Hbq1a* (hemoglobin, theta 1A), oxidative stress; *Hopx* (HOP homeobox), neural stem cells; *LOC547349* (also known as H2-K1, MHC class I family member), self-recognition; *Proser2* (proline and serine rich 2), unknown (tumor marker); *Rnmt* (RNA (guanine-7-)methyltransferase), 5′-cap structure of mRNA; *Sapcd2* (suppressor APC domain containing 2), migration and spindle orientation; *Stk38* (serine/threonine kinase 38; also known as *Ndr1*), spines of dendrites, protection of retinal neurons, and mitophagy; *Trhde* (thyrotropin releasing hormone [TRH] degrading enzyme), degradation of TRH; *Vmn1r62* (vomeronasal 1 receptor 62), pheromone receptor; *U90926* (cDNA sequence U90926), putative lncRNA.

#### 3.2.6. Upregulated PC-Related Genes

We compared the gene expression profile of the Aβ + PC group with that of the Aβ group. In total, 19 of the downregulated Aβ-related genes were >2-fold upregulated by oral administration of PC, which were referred to as upregulated PC-related genes (Figure 3). Further, 14 of the upregulated EDPC-related genes overlapped with the upregulated PC-related genes, and the remaining five genes are shown in Table 8. These five genes were also upregulated by EDPC treatment, but the extent of transcript upregulation was not significant. Of the upregulated PC-related genes, *Mgat3* (mannoside acetylglucosaminyltransferase 3; also known as GnT-III) is reportedly involved in AD.

## 4. Discussion

AD is characterized by the extracellular deposition of Aβ peptide and intracellular fibril formation by hyperphosphorylated tau protein, which results in neuronal death in affected areas of the brain [1]. Aβ deposition is caused by non-physiological cleavage of APP by BACE1 (β-site amyloid precursor protein cleaving enzyme-1; also known as β-secretase) [42]. Tau, a microtubule-associated protein, regulates microtubule stability under physiological conditions. Phosphorylated tau dissociates from microtubules and destabilizes microtubule architecture, ultimately leading to neuronal dysfunction and death [44]. Along with Aβ deposition and tau fibril formation, neuroinflammation is an integral part of AD pathology [45]. Oxidative stress and mitochondrial dysfunction have also been implicated in the etiopathology of AD [46,47]. In this study, we used a mouse model i.c.v. injected with Aβ_25–35_ peptide to recapitulate the inflammatory response caused by aggregation and deposition of Aβ and the concomitant impairment of cognitive function due to neuronal death [48]. As described earlier, some of the upregulated Aβ-related genes were found to be upregulated in previous studies using AD model mice, indicating the validity of our mouse model [12,13].

In the spontaneous alternation test, i.c.v. injection of Aβ_25–35_ was confirmed to impair the cognitive performance of the mice, which was significantly ameliorated by oral administration of EDPC. DNA microarray analyses using hippocampal RNA samples revealed that i.c.v. injection of Aβ_25–35_ aberrantly changed the gene expression profile of the hippocampus. We identified 1368 upregulated (>2-fold) and 949 downregulated (<0.5-fold) Aβ-related genes. Oral EDPC administration led to the downregulation of 3 upregulated Aβ-related genes and upregulation of 32 downregulated Aβ-related genes. These 35 genes are assumed to be involved in the ameliorative effects observed in vivo.

Orally administered EDPC counteracted the downregulation of *Prnp*, which encodes PRNP [19,20]. The pathologically misfolded isoform of PRNP (PrP^Sc^) causes prion diseases (e.g., CJD), and was further shown to be involved in AD [19]. The cellular isoform of PRNP (PrP^c^) is involved in autophagy and is essential for normal neuronal function under physiological conditions [20]. It is possible that EDPC administration contributed to the restoration of neuronal function through the normalization of *Prnp* expression. A missense mutation in *Gal3st1*, one of the upregulated EDPC-related genes, has been identified as a risk factor for CJD [23].

Orally administered EDPC counteracted the downregulation of *Cct4* and *Mtap9* (*Map9*). The latter is involved in the maintenance of microtubule architecture [21,26]. As disruption of microtubule integrity is one of the characteristics of AD, it is possible that EDPC facilitated the maintenance of microtubule homeostasis, ultimately ameliorating cognitive impairment in vivo. *Pik3cg* is mainly expressed in immune cells, including microglia in the CNS. In an ischemia/recirculation mouse model, it was shown that PI3Kγ (phosphoinositide 3-kinase gamma) restrains the neurotoxic effects of microglia [27]. *Pik3cg* is also expressed in neurons, and *Pik3cg* knockout (KO) mice display symptoms typical of attention-deficit/hyperactivity disorder (ADHD) due to the loss of synaptic plasticity [28]. Agrawa et al. described that i.c.v. injection of STZ decreased the expression of the *PI3K* gene and impaired cognitive functions in rats. Intraperitoneal injection of PC was shown to increase *PI3K* expression and to alleviate cognitive decline in vivo [7].

Interestingly, orally administered EDPC counteracted the downregulation of *Figf* (*Vegfd*), which was aberrantly repressed by i.c.v. injection of Aβ_25–35_. Vascular damage is one of the characteristics of post-mortem AD brains. VEGFD is critical for dendrite maintenance and restoration of neural function in damaged brain tissues. Accordingly, nasally delivered VEGFD mimetics mitigate stroke-induced dendrite loss and brain damage in mice [22]. EDPC may have beneficial effects on the vasculature of the brain and promote the protection of neurons. We speculate that the intranasal VEDF mimetics may alleviate AD symptoms or delay the onset of AD.

Other EDPC-related genes included *Zfand5*, *Endog*, and *Hbq1a*, which are relevant to neuroinflammation, oxidative stress, and apoptosis. ZFAND5 (ZNF216) inhibits NFκB activation [31] and acts as an activator of the 26S proteasome, stimulating overall cellular protein breakdown [32]. ENDOG was found to regulate other endonucleases in programmed necrosis or apoptosis upon translocation from the mitochondria to the nucleus [35]. Hbq1a encodes a globin variant and was shown to be upregulated in an ischemia/reperfusion mouse model [36].

EDPC could ameliorate cognitive impairment in vivo, whereas PC could not. The simplest explanation for this difference is that EDPC was more easily absorbed in the gastrointestinal (GI) tract than PC, and could thus reach higher levels in different organs of the body, including the brain. This assumption was based on the fact that EDPC is a protease-digested product consisting of no less than 58% of low-molecular-weight components with a molecular weight less than 6 kDa. In the hippocampus, all upregulated and downregulated PC-related genes were also upregulated and downregulated in response to EDPC, respectively, regardless of whether the extent of the change had reached the threshold to be considered significant. Although the two PC products, i.e., PC and EDPC, appeared to have similar effects on transcriptional profiles, there was a quantitative difference in the ratio of the upregulated and/or downregulated expressions between PC-and EDPC-related genes. They include characteristic genes, *P**rnp*, *Cct4*, *Vegfd* (*Figf*), *Map9* (*Mtap9*), *Pik3cg*, *Zfand5*, *Endog*, and *Hbq1*, which are reportedly associated with AD, CJD, and/or related neurological disorders. These suggest why their in vivo efficacies were different and EDPC but not PC had an ameliorative effect on cognitive impairment in AD model mice.

*Abat* and *Brp44* (*Mpc2*) have been reported to be relevant to AD [17,18]. ABAT is involved in the salvage pathways of purine bases and their nucleosides. At the same time, ABAT catalyzes a step in the degradation process of γ-aminobutyric acid (GABA), an inhibitory neurotransmitter, and is pivotal for the maintenance of GABA concentrations and its distribution in the brain. MPC2 (BRP44) is a mitochondrial pyruvate carrier involved in the TCA cycle, which is indispensable for ATP production in mitochondria and normal cellular functions. PC appears to augment the homeostasis of GABA and normalize mitochondrial dysfunction by ameliorating aberrant expression of *Abat* and *Mpc2*. STK38, an EDPC-related gene product important for mitochondrial quality control, was shown to coordinate the execution of macroautophagy, including mitophagy [41].

*Mgat3* is overexpressed in the brains of AD patients. MGAT3 (GnT-III) protein modifies BACE1 by the addition of bisecting *N*-acetylglucosamine (GlcNAc) to *N*-linked oligosaccharides, thus stabilizing BACE1 and leading to an overproduction of pathogenic Aβ peptides [42]. In this study, i.c.v. injection of Aβ_25–35_ downregulated *Mgat3*. This discrepancy probably reflected the difference between exogenously injected peptides and endogenously cleaved APP in the pathology of AD. MGAT3 reportedly potentiates the generation of neurites (axons and dendrites) in vitro [43].

A group of blue-green algae belonging to the genus *Arthrospira*, formerly known as *Spirulina,* utilizes a phycobiliprotein, PC, to absorb light for photosynthesis. It has been known that ingested *Spirulina* and PC have a variety of beneficial effects, including neuroprotection [5]. The chromophore phycocyanobilin has been identified as an active component mediating these beneficial effects. After absorption in the GI tract, phycocyanobilin can be converted to phycocyanorubin by a ubiquitously expressed enzyme, biliverdin reductase. Phycocyanorubin is closely homologous to bilirubin and can pass the blood–brain barrier [4]. Thus, it is reasonable that orally administered PC can alter gene expression in the hippocampus. It is plausible that the antioxidative effect of PC was mainly responsible for the beneficial effects observed, such as maintaining normal gene expression profiles in the hippocampus [49], but other mechanisms cannot be excluded. For example, Li et al. proposed that PC can alter the epigenetic regulatory network in the hippocampus [8]. As mentioned above, EDPC has more favorable characteristics than PC, such as efficient absorption in the GI tract and delivery to target organs.

## 5. Conclusions

Orally administered EDPC ameliorated cognitive impairment induced by i.c.v. injected Aβ_25–35_. Nutrigenomics studies were performed by DNA microarray-based analyses of hippocampal samples, revealing that oral EDPC counteracted the aberrant expression of 35 genes, including *Prnp*, *Cct4*, *Vegfd* (*Figf*), *Map9* (*Mtap9*), *Pik3cg*, *Zfand5*, *Endog,* and *Hbq1a,* which are reportedly associated with AD, CJD and/or related neurological disorders. These results suggest that EDPC exerts beneficial effects on cognitive function in AD model mice by modifying the gene expression profile of the hippocampus.

## Figures and Tables

**Figure 1 nutrients-13-04431-f001:**
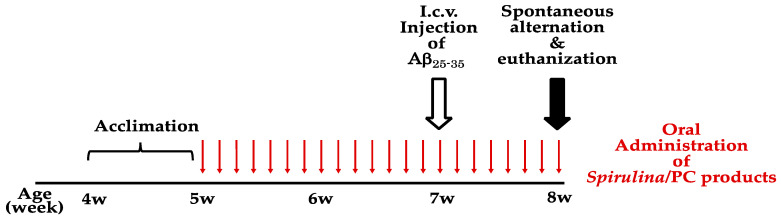
Experimental scheme. After one week of acclimation, the mice were treated with *Spirulina* (750 mg/kg), ED *Spirulina* (750 mg/kg), PC (750 mg/kg and 120 mg/kg), EDPC (750 mg/kg), or vehicle (distilled water) by daily oral administration for 22 consecutive days. At 7 weeks of age, the mice were intracerebroventricularly (i.c.v.) injected with Aβ_25–35_ (Aβ, at 10 nmol/head or vehicle. On the day of i.c.v. injection, the mice received the test substances or vehicle by oral administration more than 30 min prior to the injection. On the day of the spontaneous alternation test and euthanization, the mice were treated with the test substances or vehicle 60 min prior to the test.

**Figure 2 nutrients-13-04431-f002:**
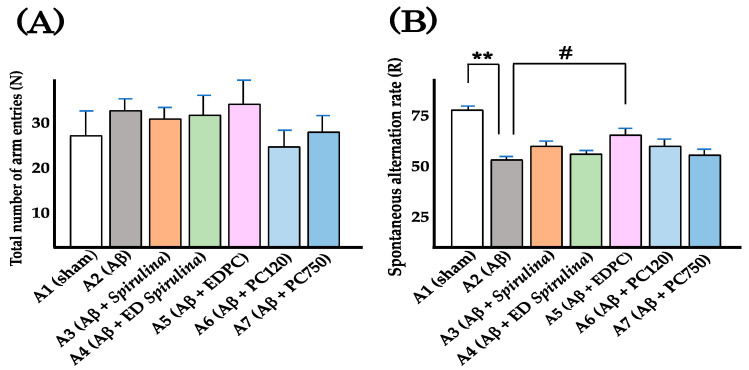
Total number of arm entries and spontaneous alternation ratio: (**A**) total number of arm entries (N). Bars and whiskers represent the mean and SEM, respectively; (**B**) spontaneous alternation ratio (R). Mice of group A6 were treated with 120 mg/kg of PC. ** *p* < 0.01, A2 (Aβ) compared with A1 (sham). # *p* < 0.05, A5 (Aβ + EDPC) compared with A2 (Aβ).

**Figure 3 nutrients-13-04431-f003:**
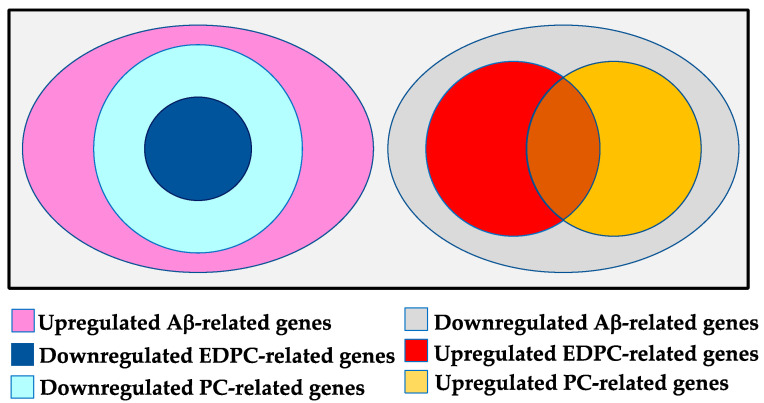
Relationship between Aβ-, EDPC-, and PC-related genes. All of the downregulated and some of the upregulated EDPC-related genes overlapped with PC-related genes. Areas are not drawn to scale.

**Table 1 nutrients-13-04431-t001:** Spontaneous alternation ratio (R) measured in a Y maze.

Group	Oral Administration	I. C. V. Injection	Total Number of Arm Entries (N)	Number of Alternation (M)	R (%) ^1^
A1	Vehicle	Vehicle	26.0 ± 5.7	17.8 ± 4.0	75.1 ± 2.5
A2	Vehicle	Aβ_25–35_	32.0 ± 2.5	15.3 ± 1.5	51.0 ± 2.1 ^2^
A3	*Spirulina*	Aβ_25–35_	30.5 ± 2.5	16.5 ± 2.0	57.2 ± 3.7
A4	ED *Spirulina*	Aβ_25–35_	31.0 ± 3.4	16.0 ± 2.5	53.9 ± 2.5
A5	EDPC	Aβ_25–35_	33.4 ± 5.7	19.2 ± 3.3	61.8 ± 3.7 ^3^
A6	PC (120 mg/kg)	Aβ_25–35_	24.5 ± 3.2	13.0 ± 2.4	56.6 ± 4.5
A7	PC (750 mg/kg)	Aβ_25–35_	26.8 ± 3.9	12.7 ± 2.2	51.6 ± 3.8

^1^ R (%) is given for each session as follows: Mx 100/(N-2). ^2^
*p* < 0.01 vs. group A1 (sham). ^3^
*p* < 0.05 vs. group A2 (Aβ).

**Table 2 nutrients-13-04431-t002:** Upregulated Aβ-related genes were also upregulated in similar AD models.

*Genes*	Gene Product		Fold Change ^1^		Reference
		Aβ ^2^	Aβ + EDPC ^3^	Aβ + PC ^4^	
*Cd9*	Cd9 antigen	3.0	3.2	4.2	[12]
*Ctsz*	Cathepsin Z	2.1	2.4	2.6	[12]
*Trem2*	Triggering receptor expressed on myeloid cells 2	3.1	2.7	3.8	[12]
*Igf1*	Insulin-like growth factor 1	3.5	2.3	3.6	[13]
*Serpine1*	Serine (or cysteine) peptidase inhibitor, clade E, member 1	4.2	4.7	3.2	[13]
*Egfr*	Epidermal growth factor receptor	3.2	2.6	3.1	[13]
*Nfe2l2*	Nuclear factor, erythroid derived, like 2 (also known as Nrf2)	3.0	2.3	3.7	[13]

^1^ Fold change with respect to the sham group (oral vehicle + i.c.v. vehicle). ^2^ The Aβ group (oral vehicle + i.c.v. Aβ). ^3^ The Aβ + EDPC group (oral EDPC + i.c.v. Aβ). ^4^ The Aβ + PC group (oral PC + i.c.v. Aβ).

**Table 3 nutrients-13-04431-t003:** Fold change values of the downregulated EDPC-related genes with respect to the sham group.

*Gene*	Product/Description		Fold Change		Function [Reference]		RPKM ^1^	
		Aβ	Aβ + EDPC	Aβ + PC		CNS E11.5	CNS E14	CNS E18
*Fbxl19*	F-box and leucine-rich repeat protein19	2.2	1.0	1.0	Ubiquitination [14]	26.6	25.5	23.7
*Pax1*	Paired box gene 1	2.9	1.2	1.4	Regulation of transcription [15]	2.4	0.4	0.0
*Zfp292*	Zinc finger protein 292	14.9	2.5	1.5	Regulation of transcription [16]	6.0	5.7	4.2

^1^ Reads per kilobase per million reads placed according to the NCBI Gene database.

**Table 4 nutrients-13-04431-t004:** Fold change values of the downregulated PC-related genes with respect to the sham group (excluding EDPC-related genes).

*Gene*	Product/Description	Aβ	Aβ + EDPC	Aβ + PC	Function [Reference]
*Abat **	4-Aminobutyrate aminotransferase	4.0	2.9	1.8	Salvage reaction Regulation of GABA [17]
*Brp44 (Mpc2) **	Mitochondrial pyruvate carrier 2	9.1	5.5	3.5	TCA cycle [18]
*Gpr123 (Adgra1)*	G protein-coupled receptor 123	7.2	4.7	3.2	Excitatory neuron
*Bsph1*	Binder of sperm protein homolog 1	2.0	1.5	0.9	Sperm
*Butr1 (Btnl10)*	Butyrophilin related 1	2.4	1.6	1.0	Putative ligand
*Casp8ap2*	Caspase 8 associated protein 2	2.3	1.5	1.1	Apoptosis
*G6pc3*	Glucose 6 phosphatase, catalytic, 3	2.3	1.4	1.0	Gluconeogenesis
*Gucy2e*	Guanylate cyclase 2e	2.3	1.5	1.0	Generation of cGMP
*Irf2bpl*	Interferon regulatory factor 2 binding protein-like	2.1	1.4	0.8	Neuronal network
*Mnt*	Max binding protein	2.1	1.5	1.0	Regulation of transcription
*Pxk (MONaKA)*	PX domain containing serine/threonine kinase	2.9	1.8	1.3	Transport of Na^+^ and K^+^
*Slx4ip*	SLX4 interacting protein	3.9	2.9	1.9	DNA repair
*Zcchc24*	Zinc finger, CCHC domain containing 24	2.1	2.0	0.6	Regulation of transcription

* Gene reportedly involved in AD.

**Table 5 nutrients-13-04431-t005:** Fold change values of the upregulated EDPC-related genes with respect to the sham group, which are reportedly relevant to AD or CJD.

*Gene*	Product/Description	Aβ	Aβ + EDPC	Aβ + PC	Function [Reference]
*Prnp*	Prion protein	0.2	1.6	2.1	Prion protein [19] Autophagy [20]
*Cct4*	Chaperonin containing Tcp1, subunit 4	0.2	1.5	2.1	Protein folding Dendrite morphogenesis [21]
*Figf (Vegfd)*	c-Fos induced growth factor	0.02	0.04	0.02	Angiogenesis Protection of neurons [22]
*Gal3st1 (Cst)*	Galactose-3-O-sulfotransferase	0.03	0.11	0.03	Myelin membrane fluidity Risk factor of CJD [23]
*Olfr181*	Olfactory receptor 181	0.29	0.73	0.54	Olfactory receptor [24]
*Olfr847*	Olfactory receptor 847	0.05	0.11	0.05	Olfactory receptor [24]
*Olfr859*	Olfactory receptor 859	0.2	1.2	1.6	Olfactory receptor [24]
*Olfr963*	Olfactory receptor 963	0.04	0.10	0.04	Olfactory receptor [24]

**Table 6 nutrients-13-04431-t006:** Fold change values of the upregulated EDPC-and PC-related genes with respect to the sham group, excluding those shown in Table 5.

*Gene*	Product/Description	Aβ	Aβ + EDPC	Aβ +PC	Function [Reference]
*Ccd163*	Coiled-coil domain containing 163	0.4	1.5	1.9	Wake [25]
*Kcnj12*	Potassium inwardly rectifying channel, subfamily J, number 12	0.2	0.8	0.7	Action potential
*Lamb4*	Laminin subunit beta 4	0.3	0.8	0.6	Laminin
*Mtap9 (Map9)*	Microtubule-associated protein 9	0.3	1.5	2.3	Microtubule Cell cycle [26]
*Pik3cg*	Phosphatidylinositol-4,5-bisphosphate 3-kinase catalytic subunit gamma	0.1	0.4	0.3	Microglia [27] Induction of ER stress, synaptic plasticity [28]
*Sema3b*	SEMA domain, immunoglobulin domain (Ig), short basic domain, secreted 3B	0.3	0.7	0.5	Removal of spines of dendrites [29]
*Tmem82*	Transmembrane protein 82	0.4	1.1	0.9	Cell proliferation [30]
*Zfand5*	Zinc finger, AN1-type domain 5	0.2	0.7	0.3	Inflammation [31] Apoptosis [32]
*Gm 14439*	Predicted gene 14439	0.1	0.1	0.2	Ribosomal protein
*LOC100503859*	RIKEN cDNA 1110015O18 gene	0.3	1.0	0.8	Unknown
*D330050I16Rik*	RIKEN cDNA D330050 gene	0.3	0.7	0.6	Unknown

**Table 7 nutrients-13-04431-t007:** Fold change values of the upregulated EDPC-related, not PC-related, genes with respect to the sham group, excluding those shown in Table 5.

*Gene*	Product/Description	Aβ	Aβ + EDPC	Aβ +PC	Function [Reference]
*Cd244*	CD244 natural killer receptor 2B4	4.0	2.9	1.8	NK cells Microglia [33]
*Chst9*	Carbohydrate (N-acetylgalactosamine4-O) sulfotransferase 9	0.03	0.11	0.05	Modification of pro-opiomelanocortin [34]
*Endog*	Endonuclease G	0.38	0.86	0.58	Apoptosis/Necrosis [35]
*Hbq1a*	Hemoglobin theta 1A	0.03	0.09	0.04	Oxidative stress [36]
*Hopx*	HOP homeobox	0.48	0.97	0.66	Neural stem cells [37]
*LOC547349*	H2-K1	0.09	0.23	0.12	Self-recognition
*Proser2*	Proline and serine rich 2	0.37	1.04	0.70	Unknown (Tumor marker)
*Rnmt*	RNA (guanine-7-)methyltransferase	0.12	0.24	0.16	5′-Cap structure of mRNA
*Sapcd2*	Suppressor APC domain containing 2	0.03	0.10	0.05	Migration Spindle [38]
*Stk38*	Serine/threonine kinase 38	0.04	0.10	0.04	Spines of dendrites [39] Retinal neurons [40] Mitophagy [41]
*Trhde*	TRH degrading enzyme	0.03	0.11	0.05	Degradation of TRH
*Vmn1r62*	Vomeronasal 1 receptor 62	0.03	0.11	0.04	Pheromone receptor
*U90926*	Putative incRNA	0.04	0.10	0.03	Putative incRNA

**Table 8 nutrients-13-04431-t008:** Fold change values of the upregulated PC-related genes with respect to the sham group (excluding EDPC-related genes).

*Gene*	Product/Description	Aβ	Aβ + EDPC	Aβ + PC	Function [Reference]
*Mgat3 **	Mannoside acetylglucosaminyltransferase 3 (GnT-III)	0.01	0.01	0.04	Modification of BACE1 [42] Neuritogenesis [43]
*Cela3b*	Chymotrypsin-like elastase family, member 3B	0.48	0.70	1.0	Protease
*Cipc*	CLOCK interacting protein, circadian	0.37	0.54	1.0	Cell cycle
*Ybx1*	Y-box binding protein 1/DNA binding protein B	0.39	0.72	0.79	Transcription and translation
*G430095P16*	Long non-coding RNA	0.08	0.13	0.17	Non-coding RNA

* Gene reportedly involved in AD.

## Data Availability

Data are available from S.N. upon reasonable request.

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
