# Peer review of "Nutrigenomic Studies on the Ameliorative Effect of Enzyme-Digested Phycocyanin in Alzheimer’s Disease Model Mice"

_nutrients, 2021, doi:10.3390/nu13124431_

Round 1
Reviewer 1 Report
The last paragraph of the Abstract is incomplete.
The 1st paragraph of the introduction contains several overstatements and needs to be revised
Reviewer 2 Report
This study covers the nutrigeomics of phycocyanins in AD mice. The writing is straightforward with appropriate theoretical approaches to reveal the effect of phycocyanins on the AD pathology. The authors mainly determine the upregulation and downregulation of the genes related to AD (and other diseases, such as CJD). This study looks attracted to the readership of the Nutrients; however, major revision of the manuscript is necessary prior to publication.
Comments:
- It may be covered by the editorial office.
- The figure caption should be located with the figure, not in the separate page.
- Please put the figures and tables in the same page when the authors mentioned them in the text.
- The authors applied Abeta(25-35) for the study. Is there scientific reason why the authors did not use major forms of Abeta, such as Abeta(1-40), Abeta(1-42), or Abeta(1-43)?
- For the experiments, the authors only inject to four-week old male mice. Please provide more information why the only male mice were applied for the experiments.
- Figure 2 shows 2 graphs indicating the ability of locomotor activity (Figure 2A) and cognition (Figure 2B).
- In the graphs, why the authors showed only (+) error bars?
- In Figure 2A, A2, A3, A4, and A5 have similar values indicating there is no significant difference between those animals but, have better locomotor activity compared to A1, A6, and A7. Then, as A2 animals were injected Abeta, the additional agents for A3, A4, and A5 may not be able to recover the damage by Abeta. Please provide this analysis in the text more detailed.
- In Figure 2B, A5 showed higher values than other experimental animals; however, is this significant?
- Please provide more information the roles/functions and/or relations to the AD (or other neurodegenerative diseases) of expressed genes. For example, why those genes are important to consider for caring/curing AD?
- In section 3.2.5, the authors suddenly mention CJD. Is there any scientific reason?
Round 2
Reviewer 2 Report
The manuscript has been revised. However, still there are some points need discussion. There are only 2 comments for the revised manuscript.
- Figure 2 shows 2 graphs indicating the ability of locomotor activity (Figure 2A) and cognition (Figure 2B).
- Based on the results presenting in Figure 2A and Table 1, the locomotor activity was not improved and the agents could recover the cognitive function. However, in Figure 2B, A5 showed higher values than other experimental animals. The authors replied to previous comments as the values are significantly higher than others. When considering the error, however, the R values are overlapped with the R values of A3 and A6. Please provide more detailed explanation about this in the text.
- My previous comment was “Please provide more information the roles/functions and/or relations to the AD (or other neurodegenerative diseases) of expressed genes. For example, why those genes are important to consider for caring/curing AD?” and the authors only revised a sentence as “EDPC may have beneficial effects on the vasculature of the brain and promote protection of neurons. We speculate that the intranasal VEDF mimetics may alleviate AD symptoms or delay the onset of AD.” Is this sentence indicating that all the genes expressed after the treatment of EDPC are related to neuroprotection? For me, this should be more clearly presented that what genes, up/down regulated by EDPC and other agents, are AD(or neuro-)-related and discussed their effects on the onset/progression of AD.
